# Non-resonant power-efficient directional Nd:YAG ceramic laser using a scattering cavity

KyeoReh Lee [1,2,4✉], Ho Jin Ma [3,4], Fabian Rotermund [1], Do Kyung Kim [3✉] & YongKeun Park [1,2✉]

Non-resonant lasers exhibit the potential for stable and consistent narrowband light sources. Furthermore, non-resonant lasers do not require well-defined optics, and thus has considerably diversified the available types of laser gain materials including powders, films, and turbid ceramics. Despite these intrinsic advantages, the practical applications of non-resonant lasers have been limited so far, mainly because of their low power efficiency and omnidirectional emission. To overcome these limitations, here we propose a light trap design for non-resonant lasers based on a spherical scattering cavity with a small entrance. Using a porous $Nd^{3+}$:YAG ceramic, directional laser emission could be observed with significant enhancements in the slope efficiency and linewidth (down to 32 pm). A theoretical model is also developed to describe and predict the operation characteristics of proposed non-resonant laser.

[1] Department of Physics, Korea Advanced Institute of Science and Technology, Daejeon 34141, Republic of Korea. [2] KAIST Institute for Health Science and Technology, Korea Advanced Institute of Science and Technology, Daejeon 34141, Republic of Korea. [3] Department of Materials Science and Engineering, Korea Advanced Institute of Science and Technology, Daejeon 34141, Republic of Korea. [4]These authors contributed equally: KyeoReh Lee, Ho Jin Ma. ✉email: kyeo@kaist.ac.kr; dkkim@kaist.ac.kr; yk.park@kaist.ac.kr

Resonators are essential components of conventional lasers (Fig. 1a), whose remarkable spatiotemporal coherence is provided by resonating modes. When no resonator is present, the gain medium is usually referred to as an amplifier to distinguish it from a laser (or an oscillator). However, a resonator-less laser was proposed by Ambartsumyan et al.[1], only 7 years after Schawlow and Townes[2] presented optical masers. To disturb the resonance, Ambartsumyan et al.[1] replaced a cavity mirror with a scattering medium (Fig. 1b). In this peculiar cavity, light does not form a closed loop as in a conventional resonator, but rather randomly wanders the cavity modes until it exits. Consequently, stable phase relations are not defined and discrete resonance frequencies no longer exist, although the intensity is still successively amplified as the round-trip progresses. This type of laser is referred to as a non-resonant laser (NRL) and distinguished from conventional lasers based on the resonant (phase-dependent) feedback. A few years later, a standalone NRL inside a scattering gain medium was also proposed[3], which was subsequently called a random laser (Fig. 1c)[4–6]. It is noteworthy that not all random lasers are NRL: several random lasers with strong scattering media are based on coherent feedback as in conventional lasers[5,6].

NRLs are highly stable against phase fluctuations and spectral instabilities. This stability is possible because, in an NRL, the lasing frequency solely depends on the intrinsic emission spectrum of a gain medium[7], which keeps the lasing frequency stable and consistent regardless of cavity configurations[8,9]. Furthermore, laser-grade optics and transparent gain media are not mandatory in NRLs, providing not only cost-effectiveness, but also opportunities to utilize a broad range of unexplored 'turbid' gain materials in the form of powders[10], films[11], and opaque ceramics[12]. NRLs have been studied as potential solutions for the frequency ranges in which optical components are not commonly available, such as the ultraviolet[13] and terahertz[14,15] regimes, and as potential multicolour lasers[16,17].

Despite the advantages above, NRLs have not been widely adopted yet. One of the major reasons is the low power efficiency. This issue originates from multiple light scattering from turbid media, which induces significant cavity loss. For example, the effective reflectance of a scattering medium reported in ref. [8] is $\sim 8.9 \times 10^{-6}$. A random laser can mitigate a cavity loss because its

cavity is surrounded by a scattering medium. However, a scattering medium also rejects the pumping light and significantly lowers the pumping efficiency[18]. The other reason is the omnidirectionality of laser emission. As a scattering medium increases the spatial degree of freedom, light is usually emitted in every possible direction in NRLs. Although such spatially incoherent features have been utilized in speckle-free imaging[19,20], this characteristic has reduced the advantages of these lasers compared to incoherent light sources such as conventional lamps and light-emitting diodes. It still remains demanding to control and narrow the laser direction without sacrificing power efficiency[15,21,22], even though power-efficient random lasers have been demonstrated[23–25].

Here we propose a power-efficient and directional NRL based on a spherical scattering cavity (Fig. 1d). The present NRL uses a light-trap design based on a small-aperture spherical cavity inside a scattering gain medium. For pumping light, the cavity works as a fish trap: light enters easily but has difficulty exiting. Most pumping light cannot escape the cavity and eventually is absorbed by the gain medium. For emission light, on the other hand, the pumped scattering gain medium amplifies the light upon each reflection, compensates the reflection loss from the wall, and prevents the emission light from extinction until the light reaches the outlet. In our method, the directionality and degree of spatial coherence can be controlled by adjusting the size of the outlet. We experimentally demonstrated the proposed concept using a porous $Nd^{3+}$:$Y_3Al_5O_{12}$ (Nd:YAG) ceramic as a scattering gain medium and achieved considerable improvements in power efficiency and directionality. A theoretical model was also introduced to explain the experimental results, to elucidate the lasing mechanism and predict the laser characteristics.

## Results

**Experimental setup.** To realize the proposed concept, we designed a unique laser system composed of two scattering gain media (Fig. 2a), each of which has a hemisphere cavity and semicylinder channel on its surface (Fig. 2b). By vertically stacking two such cavities, we established a spherical cavity with an entrance. A coupling fibre was placed through the channel to couple the cavity directly to free space (Fig. 2b). To maintain the intracavity structure, we introduced a

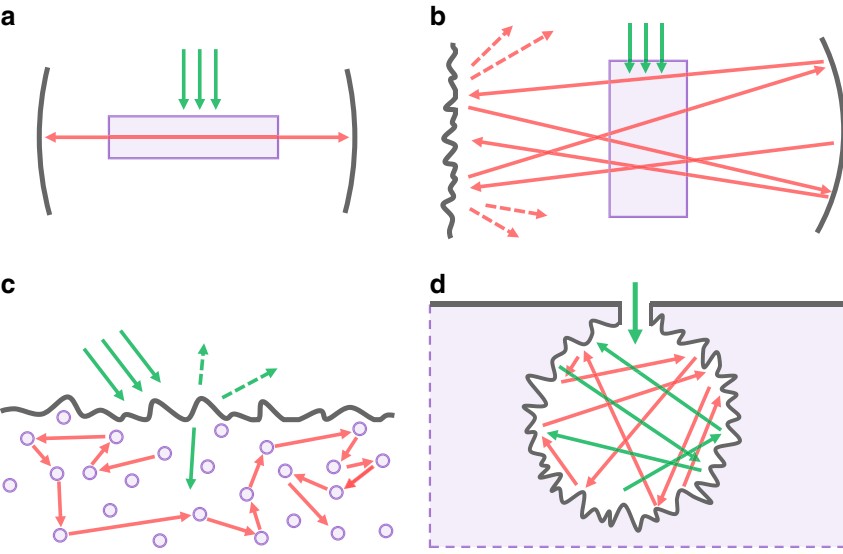

**Fig. 1 Different types of lasers. a** Conventional coherent laser with a two-mirror resonator. **b** First NRL demonstrated in ref. [1]. One mirror is replaced with the scattering medium. **c** Random laser. The scattering medium provides gain and non-resonant feedback simultaneously. **d** Proposed 'spherical light-trap' design, consisting of a non-resonant spherical scattering cavity. The scattering medium encapsulates the cavity so that light cannot escape easily. The green and red arrows indicate the pumping and emission light, respectively.

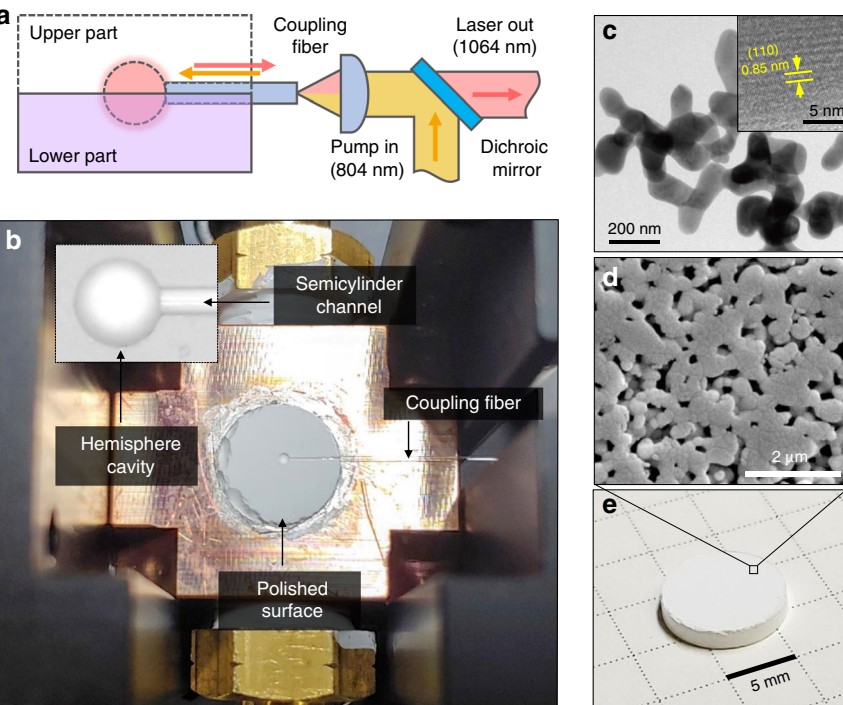

**Fig. 2 Experimental setup. a** Optical setup. See 'Methods' for details. **b** Photograph of the lower hemisphere cavity, with a cavity diameter of 800 μm and a channel diameter of 250 μm. The magnified cavity image without the fibre is shown in the inset. **c** High-resolution transmission electron microscopy (TEM) images of synthesized 1 at.% Nd:YAG nanoparticles after calcination at 1250 °C for 4 h. The inset shows the crystallinity of the particles. **d** Scanning electron microscopy (SEM) microstructure image of the Nd:YAG ceramics consolidated at 1350 °C for 10 h. **e** Porous Nd:YAG ceramic disk used as a scattering gain medium.

porous Nd:YAG ceramic material as a bulk and rigid scattering gain medium (Fig. 2c–e)[12]. In the fabrication process, we intentionally induced residual pores inside the ceramics by applying a low sintering temperature (Fig. 2d) (see 'Methods'). The estimated absorption and reduced scattering coefficients ($\mu_a$ and $\mu_s'$) are 0.00023 and 317 mm$^{-1}$ for 1064 nm and 0.02 and 356 mm$^{-1}$ for 804 nm, respectively. On the top of each disk, we engraved a hemisphere cavity with a diameter of 300, 400, 800, and 1500 μm, and a 250 μm-diameter semicylinder channel using a machining centre (Fig. 2b). The mating (top) surfaces of the disks were well polished to minimize the cavity loss through the interface (see 'Methods'). We used a bare multimode fibre with a 225-μm cladding diameter as a coupling fibre. Before stacking the two disks, the fibre was placed as shown in Fig. 2b. The internal fibre end was aligned to the cavity wall. A continuous-wave (CW) laser diode (emitting near 804 nm) was used as the pumping source. The laser output power and spectrum were measured as the pumping power increased (see 'Methods').

**Power and spectral properties**. Measured output powers for different cavity diameters including the slope efficiencies and laser thresholds are presented in Fig. 3a. It is noteworthy that the conventional random laser scheme is also tested for comparison (grey lines in Fig. 3a). Each cavity diameter is measured from a single laser device. A smooth transition rather than a sharp kink is observed at the threshold. Linear slopes are observed at above-threshold pumping power. We estimated the threshold for lasing by extrapolating the linear slope (Fig. 3a, inset). Selected normalized spectra of the 400 μm cavity are depicted in Fig. 3b. The well-known Nd:YAG emission spectrum is observable at very low pumping power (~2 mW). As the pumping power increases, ambient emission peaks are suppressed and the emission linewidth becomes very narrow. A linear redshift of the peak

wavelength is observed as the pumping power increases (Fig. 3c). The evolution of the laser linewidth is shown in Fig. 3d. The linewidth also rapidly shrinks near the threshold (notice the log scale in Fig. 3d). Above the threshold, the linewidth slowly saturates. The minimum linewidth measured is 32 pm for a 800 μm-diameter cavity, which is much narrower than the NRL bandwidths reported based on Nd$^{3+}$ crystal powders[24].

The line narrowing mechanism of NRL is based on the uneven stimulated emission cross-section [$\sigma_{st}$ in Eq. (6)] distribution in spectral domain[26], which is consistent with the line narrowing process of optical amplifiers and amplified spontaneous emission. Similar to the conventional lasers[2], we expect the effect of spontaneous emission and practical instabilities (e.g., pumping instability) should be explored to correctly anticipate the linewidths dynamics of NRLs.

The smooth transition at the threshold and its reasons are consistent with random lasers[25,27]. The sharpness of the kink at the laser threshold can be quantified by the spatial and spectral overlaps between the laser and spontaneous emission[28]. As the overlap increases, the kink becomes smoother due to the increased portion of spontaneous emission in the measured power. As our NRL design emits both laser and spontaneous emission through the coupling fibre, their spatial profile cannot be differentiated. Therefore, the observed kink is derived from the spectral narrowing and the bandpass filter we used ($\lambda_c = 1064$ nm, $\Delta\lambda_{FWHM} = 10$ nm). If our power measurement covers the entire emission spectrum range of Nd:YAG including the secondary transitions such as $^4F_{3/2} \rightarrow {}^4I_{9/2}$ (939–946 nm) and $^4F_{3/2} \rightarrow {}^4I_{13/2}$ (1319–1357 nm)[29], the kink may not be observable.

At high pumping power, reduced efficiency and linewidth broadening are observed, which can be explained by thermal effects[30]. Based on the measured redshift and ref. [30], the intracavity temperature can be estimated (Supplementary Fig. 1).

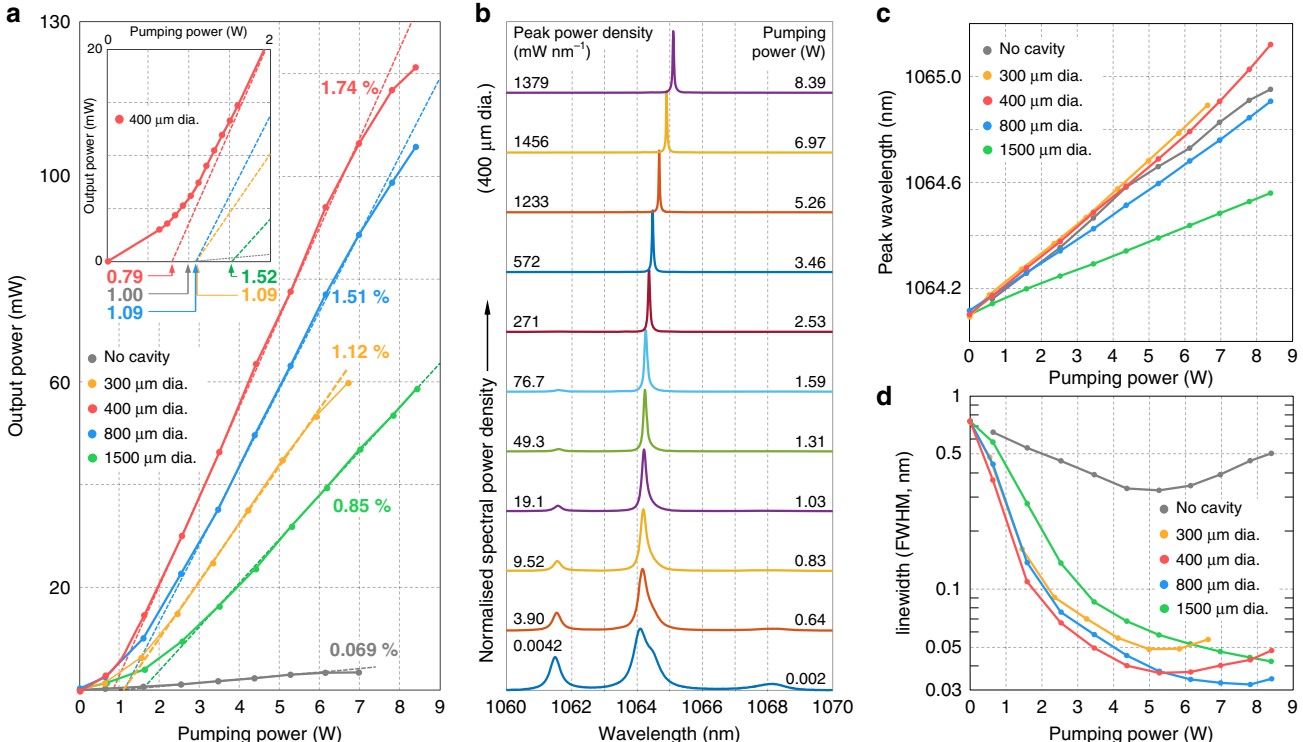

**Fig. 3 Power and spectral properties. a** Measured output powers for different cavity diameters. The inset shows the finer measurements near the threshold for a 400 μm-diameter cavity. The extrapolated thresholds are shown below. **b** Normalized spectra for a 400 μm-diameter cavity. The corresponding peak power density and pumping power are shown on the left and right sides of the peak, respectively. Note that the sampled pumping powers are not equally spaced. **c** Measured peak wavelengths for different cavity diameters. **d** Measured linewidths in terms of full width at half maximum (FWHM) for different cavity diameters. Notice the log scale of the y-axis.

The higher the cavity temperature, the greater the redshift. The highest estimated temperature is 227 °C for the peak wavelength of 1065.1 nm. Although the thermal effect is relatively common in CW pumping schemes, the temperature increase is more rapid in this case due to the localized heat sources around the small cavity.

We measured slope efficiencies of 1%–2% depending on the cavity diameter (Fig. 3a). To prove the effectiveness of our idea, a conventional random laser scheme without a cavity was also tested (see 'Methods'). Without the cavity, the most pumping light was reflected out, and a significantly lower slope efficiency (~0.07%) was observed (Fig. 3a), which clearly shows the advantage of our method. The measured laser slope efficiency seems comparable to those of random lasers using bulk ceramics[12]. For multimode lasers, however, the modal power density (or radiance) should also be explored since the integrated power does not reflect the directionality of the lasers.

The directionality of laser emission can be quantified by the number of spatial modes ($N_x$) that the laser emission occupies[31]. The smaller $N_x$ presents the more directional laser emission. The $N_x$ is calculated by multiplying the spatial and reciprocal (momentum) areas of the laser emission. For example, for a laser with an emission diameter $D$ and numerical aperture NA, $N_x = \left(\frac{\pi \mathrm{NA}}{2 \lambda} D\right)^2$, where $\lambda$ is the lasing wavelength. As we used a coupling fibre with $D = 200$ μm and NA $= 0.5$, we find $N_x \approx 21,800$ in our demonstrations, which is significantly smaller than the number of spatial modes of random lasers (for example, $N_x > 10^7$ in ref. [12]). This quantification presents that our design offers $10^2$–$10^3$ times better directionality than the existing random lasers.

**Temporal property**. To explore the temporal properties of the proposed laser, we observe the temporal dynamics of output power and spectrum. At various sampling rates and time pairs,

both fast and slow dynamics of the laser are examined separately (Fig. 4, see 'Methods' for details). We find the output power of the proposed laser is temporally stable (Fig. 4a, b). Any peculiar behaviours are not observed under the current measurement conditions. The stability is quantified by the coefficient of variation (CV, the ratio of the SD to the mean), which are 0.00037 and 0.00659 for short- and long-term measurements, respectively. The measured laser spectrum shows high temporal stability (Fig. 4a, b). Under the current measurement conditions, the laser output spectra are stable in both short- and long-term measurements. Such a high spectral stability agrees well with the previous reports on NRLs[8]. Based on the long-term spectra measurement results, we deduce the small (CV < 1%) variations in long-term power measurements are not derived from the internal laser dynamics, but from the pumping power fluctuation.

**Spatial property**. To explore the spatial properties of the proposed laser, the direct beam and 1951 USAF resolution target are imaged with and without a diffuser (see 'Methods'). In the proposed system, the spatial beam profile of the laser output is defined by the used coupling fibre and does not depend on the pumping power (Supplementary Fig. 2). For comparison, the results obtained using a conventional Nd:YAG laser are depicted as well (Fig. 5). As has been found using other NRLs[19,20] and multimode lasers[32–35], good spatial incoherency can be observed. No interference fringes from the optics and dust are apparent compared with the results using the conventional laser (Fig. 5a, b). However, Newton's rings are visible due to the multiple reflections in the protection glass of an image sensor. This result is, in fact, quite unique because the only spatially incoherent sources with sufficient coherence lengths can produce such rings. With a diffuser, our laser produces a flat intensity image, while

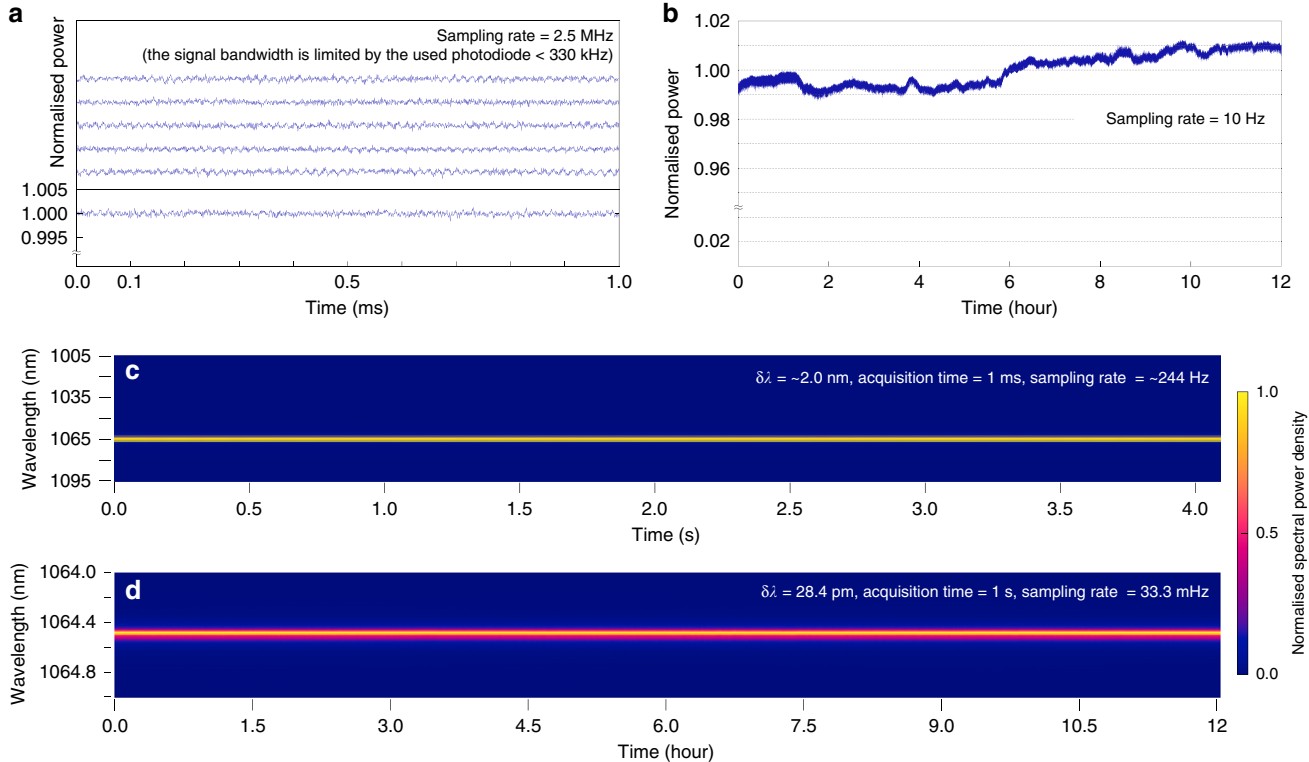

**Fig. 4 Temporal properties.** The short- (**a**) and long-term (**b**) temporal dynamics of laser power. The measured power is normalized to its mean value. All measurements in the Fig. 4a are identically scaled. The short- (**c**) and long-term (**d**) temporal dynamics of the laser spectrum. The measured spectral power density is normalized to the maximum value of the entire series. See 'Methods' for the details of the measurement setup.

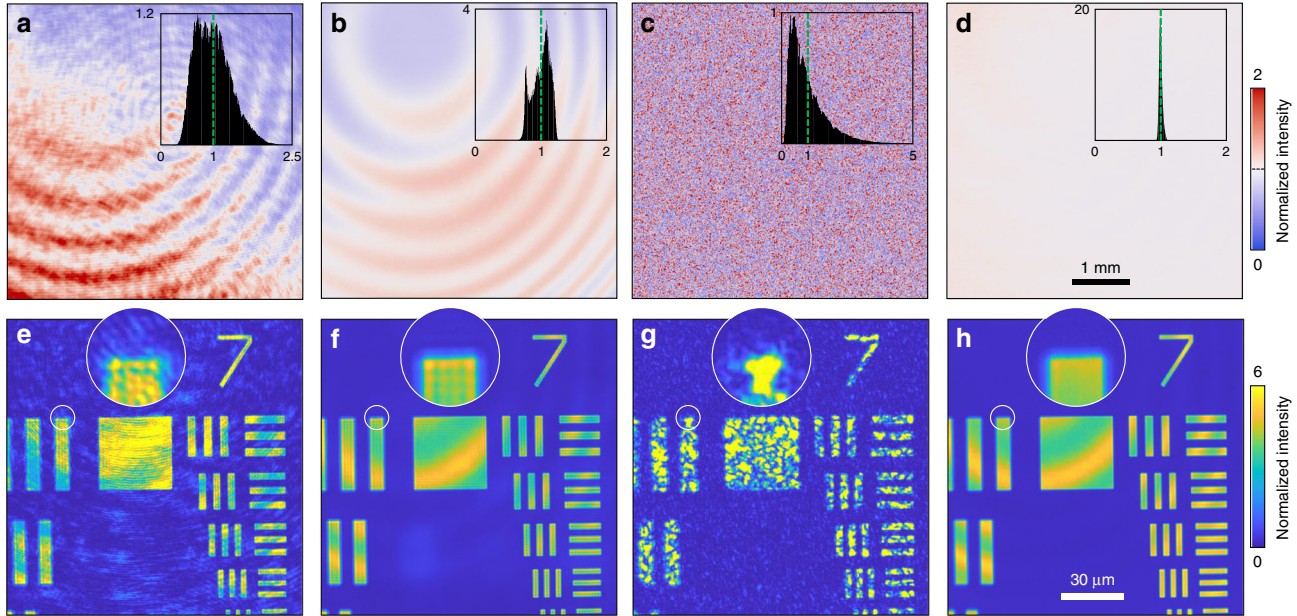

**Fig. 5 Spatial properties.** For each pair, a conventional laser (left) and the proposed laser (right) were used. **a**, **b** Direct beam images without a diffuser. **c**, **d** Direct beam images with a diffuser. The insets are the probability density functions of the corresponding images. The x-axis is normalized to the mean value (green dotted line). **e**, **f** 1951 USAF resolution target images without a diffuser. **g**, **h** 1951 USAF resolution target images with a diffuser in front of the target. See 'Methods' for the details of the optical setup.

the conventional laser exhibits a speckle pattern (Fig. 5c, d). As the diffuser provides a wide illumination angle, the rings were shifted and summed incoherently. Similar results were obtained with a USAF target. The coherent noises and speckles in conventional lasers (Fig. 5e, g) are effectively removed by using the

proposed laser, and the image contrast is significantly enhanced (Fig. 5f, h). Comparing Fig. 5f with Fig. 5h, it is noteworthy that the edge diffractions are smoothed by the diffuser (Fig. 5e–h, insets). However, the rings cannot be suppressed by the diffuser in this case due to the use of an objective lens that limits and

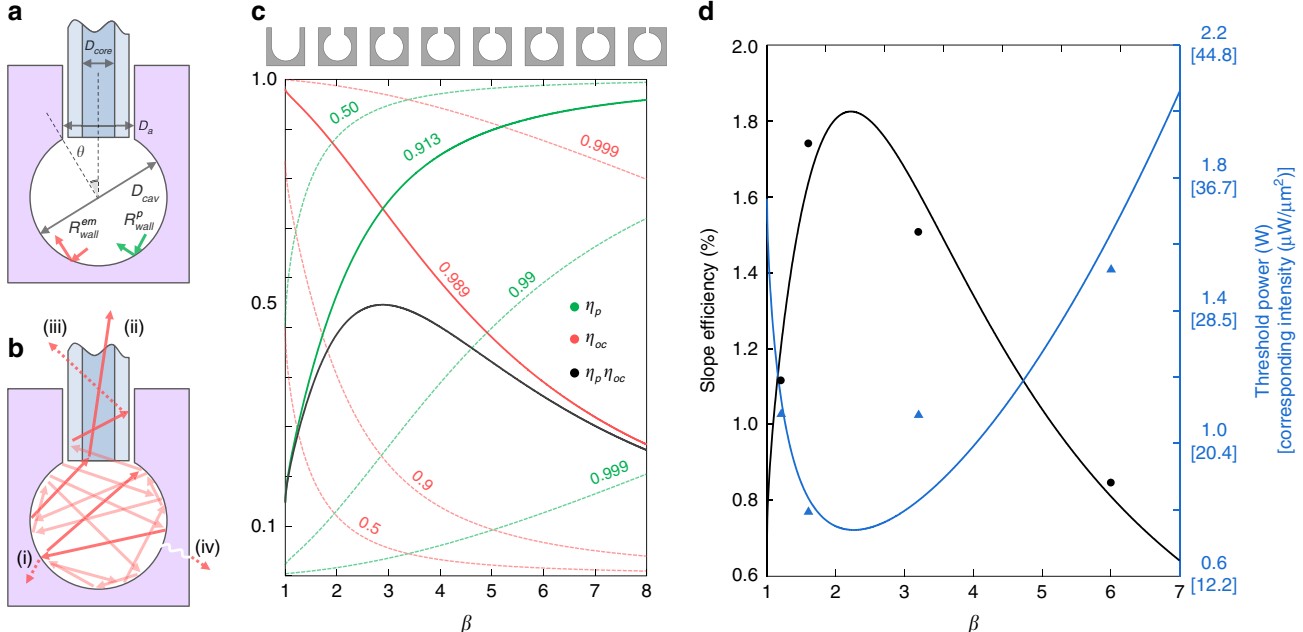

**Fig. 6 Theoretical model. a** Schematic of the theoretical model. $R_{wall}^{p}$ and $R_{wall}^{em}$ are the wall reflectances for the pumping and emission light, respectively; $D_{cav}$, $D_a$, and $D_{core}$ are the diameters of the scattering cavity, aperture, and fibre core, respectively; and $\theta = \sin^{-1}(1/\beta)$. **b** Four different origins of loss: (i) wall reflection, (ii) laser outcoupling, (iii) loss through the aperture, and (iv) undesired leaks. **c** Variations in $\eta_p$ (green lines) and $\eta_{oc}$ (red lines) with different $\beta$ as functions of $R_{wall}^{p}$ (green numbers) and $R_{wall}^{em}$ (red numbers), respectively. The schematics of cavities with different $\beta$ are provided above the graph. $\eta_p \eta_{oc}$ (black line) is shown for the experimental values of $R_{wall}^{p}$ and $R_{wall}^{em}$ (solid lines). It is noteworthy that the effects of fibre and unwanted leaks have been neglected here, i.e., $p_{noc} = p_{leak} = R_{oc} = 0$. **d** Experimental slope efficiency (left) and threshold (right) fitted by the theory, using $p_{leak} = 0.009$, $R_{noc} = 0.22$, and $\eta_0 = 0.313$. The threshold intensity is shown as well for calculation of thresholds for cavities with different sizes.

demagnifies the propagation angles (using the numerical aperture and magnification, respectively) at the image sensor plane.

**Theoretical model.** For different cavity diameters, different parameters are observed for lasing. To predict such behaviours, we developed a simple theoretical model (Fig. 6a). This model supposes a bulk and rigid scattering medium with different wall reflectances $R_{wall}^{p}$ and $R_{wall}^{em}$ for pumping and lasing frequencies, respectively. A spherical cavity of diameter $D_{cav}$ is placed inside the scattering medium with an aperture of diameter $D_a$, and a coupling fibre with a core diameter of $D_{core}$ is used.

As in conventional lasers, lasing begins when the gain exceeds the cavity loss. In our geometry, this situation occurs when the effective reflectance $R_{eff} = \sum_k p_k R_k$ becomes larger than unity, where $p_k$ and $R_k$ are the statistical probability and reflectance, respectively, of a lossy event that light can experience in the cavity. Here, we consider four possible lossy events: (i) reflection loss by the wall, $p_{wall}$; (ii) emission outcoupling, $p_{oc}$; (iii) loss through the aperture, $p_{noc}$; and (iv) undesired leaks, $p_{leak}$ (Fig. 6b). Assuming the Lambertian reflection of a scattering medium, the internal intensity distribution on the cavity wall can be considered uniform, and the statistical probabilities ($p_k$) can be estimated by the internal area fractions[36] (see 'Methods'). Subsequently, the laser rate equations can be established, and a familiar relation between the experimental pumping power $P_I$ and measured output power $P_O$ can be obtained:

$$P_O = \eta_0 \eta_p \eta_{oc} \left( \frac{\nu_{em}}{\nu_p} \right) (P_I - P_{th}), \qquad (1)$$

where $P_{th}$ is the threshold power; $\nu_p$ and $\nu_{em}$ are the pumping and emission wavelengths, respectively; and $\eta_p$, $\eta_{oc}$, and $\eta_0$ are the pumping, outcoupling, and practical efficiencies, respectively (see 'Methods'). As $\nu_{em}/\nu_p$ (or the laser quantum efficiency) and $\eta_0$ are the intrinsically and practically given constants,

respectively, the slope efficiency is a function of $\eta_p \eta_{oc}$, where $\eta_{oc} = p_{oc}(1 - R_{oc})/(1 - R_{eff}^{em})$ and $\eta_p = p_{wall}(1 - R_{wall}^{p})/(1 - R_{eff}^{p})$ are the loss fractions.

To observe the general trend, first, we plotted $\eta_p$ and $\eta_{oc}$ for different $R_{wall}^{p}$ and $R_{wall}^{em}$ (Fig. 6c) ignoring practical parameters such as reflection from the fibre ($R_{oc} = 0$), loss through aperture ($p_{noc} = 0$), and the unwanted leaks in the cavity ($p_{leak} = 0$). As the ratio of the cavity diameter to the aperture diameter, $\beta = D_{cav}/D_a$, increases, $\eta_p$ and $\eta_{oc}$ change monotonically but in opposite directions. Due to such complementary behaviours, there always exists an optimal $\beta^*$ that maximizes the slope efficiency:

$$\beta^* = \frac{1}{2} \left[ \gamma_p \gamma_{em} \left( 1 - \gamma_p \gamma_{em} \right) \right]^{-\frac{1}{2}}, \qquad (2)$$

and the corresponding $\eta_p \eta_{oc}$ is

$$\max \left( \eta_p \eta_{oc} \right) = \left( \frac{\gamma_p}{\gamma_p + \gamma_{em}} \right)^2, \qquad (3)$$

where $\gamma_p = \sqrt{1 - R_{wall}^{p}}$ and $\gamma_{em} = \sqrt{1 - R_{wall}^{em}}$. It is noteworthy again that the effect of coupling fibre and aperture loss is not considered in Eqs. (2) and (3). Especially in practical situations, the reflectances of events (ii) and (iii) in Fig. 6b are non-zero ($R_{oc}$ and $R_{noc} \neq 0$) due to the back-reflections from fibre ends and impurities, and the scattering from the cylindrical channel surfaces. Please see Supplementary Information for the more generalized equations and detailed derivations.

We found that choosing cavity-aperture diameter ratio $\beta$ properly can increase the slope efficiency greatly. For example, using the measured wall reflectances of the employed scattering medium, $R_{wall}^{p} = 0.913$ and $R_{wall}^{em} = 0.989$ (see 'Methods'), max $(\eta_p \eta_{oc})$ can reach 0.544 at the optimal $\beta^* = 2.89$, whereas it is only 0.3 at $\beta = 1.28$ or 7.01. It is worth noting that the optimized

slope efficiency, $0.544\nu_{em}/\nu_p$, is comparable to (or even greater than) that of optically pumped commercial lasers. According to Eq. (3), a higher slope efficiency is expected for higher $\gamma_p$ (or lower $R_{wall}^P$) and lower $\gamma_{em}$ (or higher $R_{wall}^{em}$).

Figure 6d shows the fits of the experimental slope efficiencies and thresholds to the proposed model. The measured wall reflectances ($R_{wall}^P = 0.913$ and $R_{wall}^{em} = 0.989$) and known properties of 1 at.% Nd:YAG[29] were utilized (see 'Methods'). The non-measurable experimental parameters such as $p_{leak}$, $R_{noc}$, and $\eta_0$ were adjusted to achieve the best fit (see Fig. 6d, caption). We found that the theoretical model could explain the experimental data with rather large practical loss $\eta_0 \approx 0.31$. We expect this loss to be due to the use of a long cylindrical channel (Fig. 2b), which would have induced practical fibre loss for both the pumping and emission light. Unlike the slope efficiency, which is independent of the physical size of the cavity, the threshold power $P_{th}$ is proportional to the surface area of the aperture (or the threshold intensity is preserved). Consequently, a smaller cavity is advantageous for the low-threshold lasers. For example, the $P_{th}$ could be decreased to ~1 mW for $D_a = 10\ \mu m$ using the same scattering gain medium. Conversely, larger cavities may require pumping thresholds that are too high for practical applications. For example, if a cavity diameter ($D_{cav}$) is increased to 10 mm, the expected pumping threshold is >200 W even for an optimized $\beta$.

Local random lasing effect on the internal wall of a cavity is not considered in this model. If local random lasing occurs, some discrepancies may exist. For example, the light may travel much longer inside the scattering medium[37], which may increase the effective reflectance and cavity lifetime, and subsequently change $\beta^*$ and $P_{th}$. Local phase-locking and laser clustering may occur[38], but global or long-range phase relations are not expected due to the disordered connections between the local lasers[39]. We expect that the macroscopic three-dimensional (3D) random geometry of our cavity will inherently prohibit the formation of global coherent modes based on the scaling theory[6,40], which makes our phase-free model work fairly well.

## Discussion

In this work, we developed and investigated a spherical 'light-trap' non-resonant scattering cavity. Enhanced power efficiency and directionality were experimentally verified, and a theoretical model was derived to elucidate and predict the lasing behaviours. The optimal ratio between aperture and cavity diameters ($\beta$) and the corresponding slope efficiency were identified based on the theory.

Stable and narrowband NRL emission can serve as a suitable frequency standard for spectroscopic applications. As sophisticated environmental control is not required to maintain cavity phase relations, we consider our NRL to be an adequate and economical alternative. For microscopy, our NRL can provide unique spatially incoherent and monochromatic (or pseudo-thermal) illumination, which is usually generated by rotating diffusers[41]. Clear self-interference patterns without coherent noise (Fig. 5) can also be advantageous in optical profilometry and polarimetry. Random distributed feedback fibre lasers may also be suitable for certain applications, as another type of NRL that provides moderate spatial coherency with pumping efficiency but requires fairly long fibres (on the order of 10–100 km)[42,43].

The degree of spatial incoherency can be adjusted by controlling the aperture area. One may either upscale or downscale the cavity to increase or decrease the number of participating spatial modes (or $N_x$). According to the theory, the same $\beta$ will yield the same slope efficiency, but the threshold will increase proportionally to the aperture area. Operation with a few spatial modes or even just one ($N_x = 1$) is possible by reducing the aperture diameter sufficiently. We expect that a spatially coherent, low-threshold, and power-efficient NRL can be achieved by properly downscaling the current approach. For smaller cavities, however, the thermal issue may worsen due to the strong heat source localization. Further study may be required for smaller designs with effective cooling systems.

Although we used the spherical cavity to simplify the theoretical approach, different cavity shapes and geometries can be beneficial for laser performance. For example, the gain medium and scattering cavity can be separated as in ref. [44] to utilize the liquid gain medium. However, since the characteristics of the proposed laser are heavily dependent on the cavity geometry, further theoretical analysis has to be applied for different geometries. In particular, the uneven intensity distribution on the intracavity surface can lead to unexpected results.

A porous and opaque Nd:YAG bulk ceramic was used as the solid-state scattering gain medium. During the fabrication of a single-crystal gain medium, the growth of defect-free and highly transparent single crystals from the melt is generally a challenging task. Low thermal conductivity and undesired inhomogeneity resulting from segregation of the incorporated dopants can also negatively affect the lasing oscillation[45,46]. For polycrystalline ceramics, many scattering elements, including residual pores, second phases, and grain boundaries, are located in the microstructure, whereby the gain medium becomes translucent or opaque[47]. We anticipate that our approach will provide a promising and unexplored alternative for gain media with difficult or lossy crystallization and consolidation processes. In addition, the proposed strategy enables these characteristics to be controlled by freely adjusting the size of the inner domains, as well as the amount, distribution, and size of the residual pores within scattering media.

We expect the proposed geometry can readily be applied to diverse scattering gain media, which has been used for random lasers. Owing to the broad applicability, our NRL can be a useful alternative platform for new lasers in the ultraviolet[13] and terahertz[14,15] frequency regions, as well as compact multicolour[16,17], efficient upconversion lasers[48,49], and stimulate the development of unexplored turbid gain materials. Laser gain material in powder or film forms can also be applied using additional spherical support. Various random lasing materials can be utilized, including ceramic powders[50,51], nanoparticles[11,17], graphene[52], and perovskites[53–55]. It is noteworthy that even and stable application of the gain material to the inner spherical wall may present a technical issue. Unfortunately, one- or two-dimensional randomly structured media utilized in random lasers may not be easy to apply the proposed geometry[15,56], as the proposed cavity always distributes the light in the internal three-dimensional space of a cavity.

## Methods

**Synthesis and characterization of Nd:YAG powders**. The co-precipitation method was used in this study to synthesize the Nd:YAG particles. Stoichiometric amounts of neodymium nitrate hexahydrate (Nd(NO$_3$)$_3$·6H$_2$O, ≥99.9%, Sigma Aldrich), yttrium nitrate tetrahydrate (Y(NO$_3$)$_3$·4H$_2$O, ≥99.99%, Sigma Aldrich), and aluminium ammonium sulfate dodecahydrate (NH$_4$Al(SO$_4$)$_2$·12H$_2$O, Reagent grade, Alfa Aesar) were homogeneously mixed in deionised water to prepare the precursor solution. The doping concentration of Nd$^{3+}$ ions was fixed at 1 at.%. The precipitant solution was prepared by dissolving ammonium bicarbonate (NH$_4$HCO$_3$, ≥99%, Sigma Aldrich) in a mixed solvent of ethyl alcohol and deionised water. The precursor solution was added drop-wise into the precipitant solution at a dripping speed of 3 ml min$^{-1}$ at room temperature. With the suspension aged for 24 h, washed repeatedly with water and alcohol in sequence, the precipitate was obtained. Then, the precursors were dried at 80 °C for 24 h in an oven. The amorphous particles underwent calcination in air at 1250 °C for 4 h to crystallize and form the garnet phase. The synthesized Nd:YAG powders were characterized using a conventional powder X-ray diffraction (XRD) technique with Cu Kα radiation at room temperature at a scan rate of 5° min$^{-1}$ between 10° and 80° (SmartLab, Rigaku). The primary particles were characterized using TEM

(Tecnai G2F30 S-Twin, FEI) (Fig. 2c). The crystallinity was found with a lattice distance of 0.850 nm, corresponding to the (110) plane of the YAG phase (Fig. 2c, inset). The XRD results agree well with the expected cubic YAG phase (JCPDS #33-0040, Supplementary Fig. 3), confirming that no impurities or secondary phases were induced during the incorporation of $Nd^{3+}$.

**Fabrication of Nd:YAG porous bulk ceramic**. As a next step, the powders were uniaxially dry-pressed under 20 MPa into a 10 mm-diameter stainless steel mould with a sample thickness of 2 mm. To prepare the porous Nd:YAG pellets, the green bodies were fired at 1350 °C using a heating rate of 5 °C min$^{-1}$ in air. The samples were consolidated for 10 h. The averaged dimensions of the ceramic disks after sintering were ∅ $8.87 \times 1.55$ mm$^2$. The microstructure images of densified Nd:YAG ceramics were obtained by scanning electron microscopy (Model Philips XL 30 FEG, Philips) after polishing to 1 µm using a diamond slurry (Fig. 2d). Many residual pores were found, as the sintering temperature (1350 °C) was insufficient for applying transparent ceramics. Notably, the remnant pores that served as a scattering factor were uniformly distributed. The average grain size of the consolidated sample was determined by multiplying the linear-intercept length of 300 grains times 1.56[57,58]. The bulk density was measured using Archimedes' method. We achieved a fractional density of 75% with an average domain size of 400 nm.

The diffuse reflectance was measured in a wavelength range of 300–1200 nm (SolidSpec-3700, Shimadzu). Low reflectance was observed for known absorption peaks of Nd:YAG (Supplementary Fig. 4). A reflective BaSO$_4$ plate was used as a reference. We used the inverse adding-doubling method to estimate the absorption and scattering coefficients of scattering media[59]. We prepared a thinner (~1 mm) porous Nd:YAG ceramic disk specimen to increase the accuracy of the transmittance measurements. We used a coherent Nd:YAG laser ($\lambda = 1064$ nm, MATRIX 1064 CW, Coherent, Inc.) and laser diode ($\lambda \approx 804$ nm, D8010C1FX, Laserglow Technologies) in the measurements. Following the instructions provided by Prahl[59], we carefully measured the reflectance and transmittance using an integrating sphere (IS236A, Thorlabs, Inc.). A commercial reflectance standard (SRS-99-010, Labsphere, Inc.) was used as a 99% reflectance reference. The estimated absorption and reduced scattering coefficients ($\mu_a$ and $\mu'_s$) are 0.00023 and 317 mm$^{-1}$ for 1064 nm and 0.02 and 356 mm$^{-1}$ for 804 nm, respectively.

**Engraving cavity structures**. Before engraving, we attached the ceramic disks to the copper holders ($25 \times 20 \times 20$ mm$^3$) with circular potholes (∅ $10 \times 1$ mm$^2$) for the mounting. The copper holders have a cylindrical channel (∅ 6 mm) under the disks for water cooling. The closest distance between the water channel and the ceramic disk is 2 mm. Indium (99.99%, IN-203321, The Nilaco Corp.) was used to fill the residual volumes of the potholes and to hold the disks firmly. We placed the assembly into an oven (250 °C, 10 min) to reinforce the holding ability of indium by melting and hardening. On two surfaces of ceramic, hemisphere cavities and semicylinder channels were machined. The centre position was set based on the edges of the copper holders. Diamond-coated micro-sized ball end mills were used. In the engraving process, an excess thickness (20 µm) was also introduced for polishing. The top surfaces of the disks were polished using coarse (6 µm grit, LF6D, Thorlabs, Inc.) and fine (1 µm grit, LF1D, Thorlabs, Inc.) polishing films in order.

**Optical system setup**. A bare multimode fibre (NA = 0.5, core/cladding diameters = $200 \pm 5$ µm/$225 \pm 5$ µm, FP200ERT, Thorlabs, Inc.) was used for the cavity coupling fibre. The fibre coating was carefully stripped using a T10S13 (Thorlabs, Inc.). The internal fibre end was ground by coarse lapping film (30 µm grit, LF30D, Thorlabs, Inc.) to prevent unwanted focusing of pumping light in the cavity, while the other fibre end was cleaved well (S90R, Thorlabs, Inc.). The internal 1 mm-range cladding was removed to prevent melting (the cladding of the used fibre was composed of hard polymer) due to the high cavity temperature during laser operation. The heat was dissipated by deionized water circulation through the upper and lower copper units set temperature for the chiller (VTRC-620, Jeio Tech. Co., Ltd) is 19 °C. An aspheric lens ($f = 4.51$ mm, NA = 0.55, C230TME-B, Thorlabs, Inc.) was used to couple the pumping light to the fibre. We employed an 804-nm-centred laser diode as the pumping source (>10 W, D8010C1FX, Laserglow Technologies). We calibrated the pumping power after the fibre using a calibrated integrating sphere (IS236A, Thorlabs, Inc.). A dichroic mirror (FF875-Di01-25 × 36, Semrock) and emission filter ($\lambda_c = 1064$ nm, $\Delta\lambda_{FWHM} = 10$ nm, FL1064-10, Thorlabs, Inc.) were used to separate the laser emission from the pumping light. The commercial silicon photodiode (S121C, Thorlabs, Inc.) and optical spectrum analyser ($\delta f = 7.5$ GHz, $\delta\lambda = 28.4$ pm, at 1064 nm, OSA201C, Thorlabs, Inc.) were used for the power and spectrum measurements, respectively. A conventional random laser scheme was also tested. The surface of the same Nd:YAG ceramic was used instead of a cavity structure. Identical fibres were employed to deliver and collect the pumping and emission light. The surface of the gain medium was aligned perpendicular to the fibre. The axial position of the fibre end was set to touch the surface slightly to maximize the intensity on the gain medium surface and to avoid bending.

For temporal dynamic of laser power, we used a commercial photodiode (SM05PD1A, Thorlabs, Inc.) with an amplifier (PDA200C, Thorlabs, Inc.). Due to the moderately high load resistance of the amplifier, the bandwidth of the

photodiode can be calculated as <330 kHz. For the short-term observation, an oscilloscope (TDS2004C, Tektronix, Inc.) is used. The time traces of 1 ms are measured at the sampling rate of 2.5 MHz. For the long-term observation, a data acquisition board (NI USB-6229, National Instruments Corp.) is used. To prevent the aliasing effect, we used a passive low-pass filter (EF502, Thorlabs, Inc.). We acquired the time trace of 100 ms with the acquisition rate of 250 kHz and take the averaged value to prevent the memory overflow and aliasing. The effective framerate is 10 Hz. The 1500 µm-diameter cavity laser is used.

For measuring the temporal dynamics of laser spectra, we used different apparatus for different time scales. For the short-term observation, a near-infra-red spectrometer ($\delta\lambda = \sim 2.0$ nm, NIRQUEST + 1.7, Ocean Optics, Inc.) is used. We used the acquisition time of 1 ms with the frame time of $4.1 \pm 0.062$ ms (mean ± SD), or the framerate of ~244 Hz. For the long-term observation, the optical spectrum analyser used in Fig. 3b ($\delta f = 7.5$ GHz, $\delta\lambda = 28.4$ pm, OSA201C, Thorlabs, Inc.) was utilized again. A laser spectrum is acquired in every 30 s for 12 hours. The 1500 µm-diameter cavity laser is used.

To explore the spatial properties of the laser, a commercial complementary metal–oxide–semiconductor camera for visible light (FL3-U3-13Y3M-C, FLIR® Systems, Inc.) was utilized. We placed an additional beam collimator ($f = 300$ mm, LA1256-B, Thorlabs, Inc.) and formed a 4f configuration with the aspheric lens. In this experiment, we used a 400 µm-diameter cavity. For comparison, a conventional Nd:YAG laser ($\lambda = 1064$ nm, MATRIX 1064 CW, Coherent, Inc.) was also tested. We spatially filtered and magnified the laser beam to make the beam cover the entire camera sensor. To obtain Fig. 5a–d, the camera directly captured the incoming beams with and without an optical diffuser (60°, 54–505, Edmund Optics Inc.). An iris was placed before the diffuser to make the speckle grain size larger than the camera pixel size (4.8 µm). We adjusted the aperture size to make the speckle grains larger than the camera pixels (4.8 µm). To produce Fig. 5e–h, a microscopic imaging system was configured using a near-IR objective lens (NA = 0.85, ×100, LCPLN100XIR, Olympus Crop.) and tube lens ($f = 60$ mm). The expected total magnification was ×33.3. A negative 1951 USAF resolution target (38–256, Edmund Optics Inc.) was imaged, with and without the diffuser before the target.

**Probability calculation**. As shown in Fig. 6b, we assumed that light could experience four different lossy events in the cavity. A proper probability $p_k$ should be set for each event to predict the lasing properties correctly, such as the pumping and outcoupling efficiencies, $\eta_p$ and $\eta_{oc}$, respectively. As the statistical probabilities are closely related to the steady-state intensity distribution on internal cavity walls, we first need to clarify the intensity distribution to correctly estimate the statistical probability and laser efficiencies.

For the intensity distribution in the scattering cavity, we adopted the calculations, which have used for integrating spheres[36]. By assuming the Lambertian reflections, the probability that light travels from an arbitrary point to the other arbitrary points on the internal surfaces of a spherical scattering cavity does not depend on the incident and scattered angle of light, and the internal intensity distribution becomes uniform. Still, the initial spatial profile of pumping light may become dominant as the internal area fraction of aperture increases or the wall reflectivity for pumping light decreases. To minimize such effects, we distributed the pumping light to the cavity surface by coarsely grinding the internal fibre end.

Under the uniform intensity distribution, we can calculate the statistical probabilities $p_k$ simply by the internal area fractions of the wall and aperture (Fig. 6b)[36]. The total inner surface area was composed of the spherical wall area, $A_{wall} = \frac{\pi}{2}D_{cav}^2(1 + \cos\theta)$ and an aperture area $A_a = \frac{\pi}{4}D_a^2$, where $\sin\theta = D_a/D_{cav}$ (Fig. 6a). In this case, the area fractions become

$$p_{wall} + p_{leak} = \frac{A_{wall}}{A_{wall} + A_a} = \frac{2}{3 - \cos\theta}, \qquad (4)$$

$$p_{oc} + p_{noc} = \frac{A_a}{A_{wall} + A_a} = \frac{1 - \cos\theta}{3 - \cos\theta}. \qquad (5)$$

As the $p_{leak}$ is a practical parameter, the $p_{wall}$ can be calculated from the estimated value of $p_{leak}$. The $p_{oc}$ and $p_{noc}$ can be estimated by calculating the coupling efficiency of coupling fibre (Fig. 6a),

$$p_{oc} = \left(NA\frac{D_{core}}{D_a}\right)^2\frac{1 - \cos\theta}{3 - \cos\theta} \quad \text{and} \quad p_{noc} = \left[1 - \left(NA\frac{D_{core}}{D_a}\right)^2\right]\frac{1 - \cos\theta}{3 - \cos\theta} \qquad (6)$$

where $D_{core}$ and NA are the core diameter and numerical aperture of the fibre, respectively.

**Laser rate equations**. The rate equations of the proposed model for the population inversion $N$ and number of photons $\phi$ are almost identical to the conventional ones[60]:

$$\frac{dN}{dt} = R_p - \frac{\sigma_{st}c}{V_\phi}\phi N - \frac{1}{\tau_2}N. \qquad (7)$$

$$\frac{d\phi}{dt} = \left( \frac{\sigma_{st} c}{V_\phi} V_N N - \frac{1}{\tau_{cav}} \right) \phi, \qquad (8)$$

where $N$ is the population inversion, $\phi$ is the number of photons, $\sigma_{st}$ is the stimulated emission cross-section, $R_p$ is the pumping rate, $\tau_2$ is the effective lifetime of upper laser level, $\tau_{cav}$ is the cavity lifetime, $V_N$ is the mode volume in the active medium, $V_\phi$ is the mode volume in the cavity, and $c$ is the speed of light. The four-level laser scheme is assumed. Since NRLs do not have 'mode' volumes, the effective 3D volume of the photon distribution should be used instead for $V_\phi$ and $V_N$. From the stationary solutions of Eqs. (7) and (8), Eq. (1) can be derived with the pumping threshold:

$$P_{th} = \frac{1}{\eta_0^p \eta_p} \frac{1}{\tau_{cav} \tau_2} \frac{h\nu_p}{\sigma_{st} c} V_\phi, \qquad (9)$$

where $\eta_p$ is the cavity pumping efficiency in Eq. (1) and $\eta_0^p$ is the practical efficiency of the pumping process. It is noteworthy that $\eta_o = \eta_0^p \eta_0^{em}$ in Eq. (1) is the total practical efficiency that combines the imperfections in both the pumping and emission processes, $\eta_0^p$ and $\eta_0^{em}$, respectively.

According to Fig. 6a, we estimated $V_\phi$ as the sphere volume without the spherical cap with polar angle $\theta$:

$$V_\phi = \frac{\pi}{24} (D_{cav} + 2\delta D_{cav})^3 (2 - \cos\theta)(1 + \cos\theta)^2, \qquad (10)$$

where $\theta = \sin^{-1}(1/\beta)$ and $\delta D_{cav}$ is the effective penetration depth of the emission light for the used scattering medium. For $\tau_{cav}$, the lifetime for the integrating spheres is used[61]:

$$\tau_{cav} = -\frac{1}{\ln R_{eff}} \left( \frac{\bar{d}}{c} + \delta\tau_{cav} \right), \qquad (11)$$

where $\bar{d}$ is the mean distance between wall reflections and $\delta\tau_{cav}$ is the effective dwell time of the emission light for the used scattering medium. We numerically calculated $\bar{d}$ (see Supplementary Fig. 5). It is worth noting that $\bar{d}(\theta<1) \approx \bar{d}(\theta = 0) = 2/3 D_{cav}$. The effective dwell time and penetration depth of the used scattering medium were calculated based on the measured scattering and absorption coefficients[62]. Specular reflection between the air and scattering medium interface was also considered roughly using the effective refractive index and measured volume fraction of the employed scattering medium. The detailed equations can be found in the Supplementary Information. In this work, both $\delta D_{cav}/D_{cav}$ and $\delta\tau_{cav}/\tau_{cav}$ were less than 0.05. The known properties of 1% Nd:YAG were utilized, $\tau_2 = 230\,\mu s$ and $\sigma_{st} = 6.5 \times 10^{-23}\,m^2$ from ref. [29].

## Data availability
The data that support the findings of this study are available from the corresponding authors upon reasonable request.

## Code availability
The MATLAB codes for the theoretical analysis in Fig. 6 are available here: https://doi.org/10.5281/zenodo.4013483.

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

## Acknowledgements

This work was supported by KAIST Up programme, BK21+ programme, KAIST Advanced Institute for Science-X, Tomocube, National Research Foundation of Korea (NRF) (2017M3C1A3013923, 2015R1A3A2066550, 2018K000396, and 2018R1A6A3A01011043). K.L. thanks Yong-Jae Kim (Samwoo Tech Co., Ltd, Daejeon, Republic of Korea) for machining centre operation on the ceramics based on our design. D.K. thanks to the Ministry of Trade, Industry & Energy (MOTIE) of Korea (PN: 10047010) for support.

## Author contributions

H.M. synthesized and fabricated the Nd:YAG scattering media. H.M. and K.L. measured properties of the scattering media. K.L. developed the theory, performed the experiments, and analysed the data. F.R., D.K., and Y.P. supervised the project. All authors wrote the manuscript.

## Competing interests

The authors declare no competing interests.
