## [Peer Review File · Nature Communications]

REVIEWER COMMENTS

Reviewer #1 (Remarks to the Author):

I have read the manuscript with an interest. The authors present nice experimental results on random Nd:YAG ceramic laser demonstrating an advanced laser design. In contrast to previously reported random lasers, the new laser cavity employs a spherical scattering cavity with a small entrance that operates as a light mode mixer, light scatter and amplifier simultaneously. The manuscript is clearly written, well referencing, and contains a good amount of experimental work. In my opinion, it deserves publication in Nature Communications after mandatory revision by the authors to meet the following concerns:

1. The authors use the term "nonresonance feedback" that traditionally relates to the lasers with a spherical cavity without internal amplification. Could you clarify the difference between the nonresonance feedback lasers and random lasers, if any?
2. The authors do not discuss temporal laser dynamics. What are the typical time traces of the laser radiation recorded with different time resolutions? Are any spikes observed?
3. The same concerning the recorded optical spectra. What is the averaging time? What is the laser spectrum performance without averaging? Are any spikes?
4. What is the mechanism of the laser line narrowing with the pump increase? Could the proposed model give an explanation?
5. What is the spatial profile of the output laser radiation? is it sensitive to the pump level?
6. It would be nice to illustrate the steady-state spatial distribution of the pump and laser power over the sphere surface estimated by the model.
7. The reflection from the fiber face is not taken into account for the results shown in Fig. 5. Could you please comment importance of this effect? Does this reflection higher or lower than the reflection from the cavity walls? What is the mechanism preventing intracavity lasing?

Reviewer #2 (Remarks to the Author):

Report on "Non-resonant power-efficient directional Nd:YAG ceramic laser using scattering cavity" by KyeoReh Lee et al.

In this paper, KR Lee et al. investigate non-resonant laser based on a spherical cavity with a small entrance. Using a small-aperture spherical cavity inside porous ND₃₊: YAG ceramic medium, laser-like emission is observed with significant enhancement in the slop efficiency and linewidth. By adjusting the size of the outlet and diameter of the spherical cavity, the emission properties of the system were tuned. A theoretical model was also developed to elucidate and predict the lasing behaviors of the proposed non-resonant laser. The results seem to be correct, and they are clearly presented. There is no doubt that the paper deserves to be published. However, I am not sure that there is enough novelty to meet the publication criteria for Nature Communications. The presented results are somewhat expected. I also find some important details are missing in the manuscript.

1. The author claims that the proposed non-resonant light trap design leads to power-efficient directional lasing emission. However, in the manuscript, no information about the emission direction is provided.
2. The whole paragraph on page 5, "The smooth transition at the threshold ... effectively lifts the power efficiency" is unclear.
3. The cooling scheme to reduce the thermal effects is not properly discussed in the manuscript.
4. Figures 4c and 4c are not referred to in the text.
5. It is not clear how the scattering gain medium amplifies the light upon each reflection and prevents extinction.
6. Some crucial information like the scattering mean free path of the porous ceramic medium, the penetration depth of the pump light, and the surface roughness of the inner wall of the cavity are missing in the manuscript.

Reviewer #3 (Remarks to the Author):

A random laser does not necessarily need a cavity, but some designs have one. Here is an example:
<https://www.nature.com/articles/srep35225>

In that design, a gain medium is surrounded by a scattering surface (by the way, I think that it would be appropriate to cite that paper). As far as I understand, the main idea behind the manuscript I'm reviewing is to invert that concept and make an empty cavity surrounded by a gain medium, where the cavity surface scatters light. A fiber couples the pump light into this hollow cavity and also couples the emitted light out of it. As far as I know, this is a new idea. Unfortunately, I do not have the necessary background to judge how good the idea is. Neither the slope efficiencies nor the pumping thresholds reported here look outstanding to me, but I am not an expert in random lasers, so I do not want to draw any conclusions from this fact. I would just like to encourage the authors to clarify whether their design outperforms other random lasers in any particular respect. Whether it does or does not, it should be written explicitly.

My biggest problem with this manuscript is its theoretical part. I find it so incomplete that I cannot fully understand what the authors did. I think I understand the concept of effective reflectance for the pump light -- it must be the ratio of the power that gets reflected back into the fiber to the power that is sent into the fiber. But what is the effective reflectance for the emitted light? Also, I do not understand Eq. (S2) even for the pump light. If a pump photon enters the cavity via the fiber, it will get reflected from a wall for sure. Then the probability that it will enter the fiber again depends on what happens next: whether the photon gets into the surrounding gain medium or hits the wall again at another place or hits the fiber. I just do not see how Eq. (S2) reflects all this. If Eq. (S2) is correct, the authors must explain it in a detailed and careful way, so that even a reader like me, who apparently lacks some relevant knowledge, can understand the very basics of their model. The same applies to Eq. (S1).

The authors highlight the perspective of new lasers in the ultraviolet and terahertz frequency regions as potential further developments based on their approach. Here, I find that the authors owe the community a couple of simple estimations in the Supplementary Materials. Based on their theoretical understanding, do they have any evidence that their design offers advantages for ultraviolet or terahertz lasing, as compared to other random lasers?

A few minor remarks:

- I think that the first equation on line 419 is wrong -- the first denominator should be 4 instead of 2.
- Please check what you wrote on line 179.
- In Figure 2b, I do not see any "semicylinder channel" with a diameter of 250 micrometers.

In summary, my overall impression of this manuscript is positive, I find its main idea novel and appealing, and if the authors can highlight in what respect their approach outperforms other random lasers, I recommend the manuscript for Nature Communications. Otherwise, if it is just a nice idea that has some potential, then a journal like Scientific Reports is probably better suited for disseminating these results. In any case, the authors should heavily revise their manuscript for clarity, especially its theory part.

Reviewer #1 (Remarks to the Author):

I have read the manuscript with an interest. The authors present nice experimental results on random Nd:YAG ceramic laser demonstrating an advanced laser design. In contrast to previously reported random lasers, the new laser cavity employs a spherical scattering cavity with a small entrance that operates as a light mode mixer, light scatter and amplifier simultaneously. The manuscript is clearly written, well referencing, and contains a good amount of experimental work. In my opinion, it deserves publication in Nature Communications after mandatory revision by the authors to meet the following concerns:

We sincerely appreciate the careful reading and constructive comments.

1. The authors use the term "nonresonance feedback" that traditionally relates to the lasers with a spherical cavity without internal amplification. Could you clarify the difference between the nonresonance feedback lasers and random lasers, if any?

Thank you for the comment. Most random lasers can be understood as non-resonant feedback lasers, while a few exceptional random lasers are based on resonant feedback. For more information, please see [Cao, Hui. *Waves in random media* 13.3 (2003): R1-R39; Cao, Hui, et al. *IEEE Journal of selected topics in quantum electronics* 9.1 (2003): 111-119].

A random laser with *non-resonant* feedback operates when its pumping power is sufficient enough to amplify one or more photons before a photon leaves a cavity (that is, the scattering media for random lasers). Because the principles of non-resonant random lasers do not limit the degree of scattering of a gain medium [Letokhov, V. S. *Sov. Phys. JETP* 26 (1968): 835-840.], strong pumping power can initiate the laser even in a weak scattering medium such as human tissue [Polson, Randal C., and Z. Vally Vardeny. *Applied physics letters* 85.7 (2004): 1289-1291]. Due to its ease of implementation, the majority of the previously reported random lasers are based on non-resonant feedback.

A random laser with *resonant* feedback operates when a photon forms a closed-loop path in a scattering medium. In this case, the scattering medium serves as a laser oscillator. Therefore, very sharp peaks on emission spectrums are usually observed in a random laser with resonant feedback, as in conventional lasers [Cao, H., et al. *Applied Physics Letters* 76.21 (2000): 2997-2999]. However, to form a closed-loop in a scattering medium, extremely strong scattering is required that satisfies $l_t \approx \lambda$, where the l_t is the transport mean free path of the scattering medium and λ is a lasing wavelength. Therefore, the random laser with coherent feedback is hard to realize, particularly for 3-D. Please note that 1-D or 2-D random lasers with coherent feedback are relatively easy to demonstrate, for example, [Abaie, Behnam, et al. *Light: Science & Applications* 6.8 (2017): e17041-e17041].

As suggested, we clarified the difference between non-resonant feedback lasers and random lasers in the introduction of the revised manuscript.

2. The authors do not discuss temporal laser dynamics. What are the typical time traces of the laser radiation recorded with different time resolutions? Are any spikes observed?

We appreciate the constructive comment. As the reviewer suggested, we performed additional experiments and measured the time trace data of the laser emission in different sampling rates and time scales. To observe short- and long-term temporal variations, we measured the time trace of 1 ms and 12 hours with acquisition rates of 2.5 MHz and 10 Hz, respectively (Figs. R1a and R1b). In short-term measurements, randomly fluctuating power is observed. We do not find any odd features such as spikes. In a long-term measurement, slow power drifting is observed in addition to the random fluctuations. Based on the long-term spectrum measurement result (Fig. R1d), we deduce the variations in long-term power measurement is not derived from the internal laser dynamics, but from the pumping power fluctuation. We calculated the coefficient of variation (CV, the standard deviation divided by the mean) to quantify the degree of laser power fluctuation. In short and long-term measurements, CV = 0.00037 and 0.00659, respectively. As suggested, the time traces are added (Fig. 4 of the revised manuscript) and discussed in the revised manuscript.

Figure R1. Temporal properties. **a** and **b**, the short- (**a**) and long-term (**b**) temporal dynamics of laser power. The measured power is normalized to its mean value. All measurements in figure 4a are identically scaled. **c** and **d**, the short- (**c**) and long-term (**d**) temporal dynamics of the laser spectrum. The measured spectral power density is normalized to the maximum value of the entire series. See Methods for the details of the measurement setup.

3. The same concerning the recorded optical spectra. What is the averaging time? What is the laser spectrum performance without averaging? Are any spikes?

We utilized an optical spectrum analyzer (OSA) to measure the optical spectra in Fig. 3b, and a long acquisition time (>1 s) is required to retrieve a single spectrum.

We agree that the temporal evolution of a laser spectrum is an important specification, and thus we observed the time-resolved laser spectrum in various temporal scales. Similar to the power measurements, the measurements for short- and long-term temporal variations were performed. For the short-term measurement, we used a grating-based spectrometer ($\delta\lambda = \sim 2.0$ nm, NIRQUEST+1.7, Ocean Optics, Inc.) with the acquisition time of 1 ms and the frame time of 4.1 ± 0.062 ms (mean \pm standard deviation, or the framerate of ~ 244 Hz). A temporal trace of the spectra was acquired for 4 seconds (1,000 sampling points). For the long-term measurement, we also used the OSA. A laser spectrum is acquired in every 30 seconds for 12 hours. The measured laser spectrum shows high temporal stability (Figs. 4a and 4b). Under the current measurement conditions, the laser output spectra exhibit static behaviors in both short- and long-term measurements. Such a high spectral stability agrees well with the previous reports such as [Ambartsumyan, R., et al., *Sov. Phys. JETP*, 1967. 24: p. 481-485]. As the reviewer suggested, the time-resolved spectrum results are added and discussed in the revised manuscript.

4. What is the mechanism of the laser line narrowing with the pump increase? Could the proposed model give an explanation?

We appreciate the constructive question. According to [Ambartsumyan, R. V., et al. *JETP* 24 (1967): 1129], the bandwidth narrowing mechanism of a non-resonant laser (NRL) is due to the uneven stimulated emission cross-section [σ_{st} in equation (7)] in the spectral domain, which is consistent with that of optical amplifiers and amplified spontaneous emission (ASE). Because the cavity of an NRL does not provide any phase relations (or mode) on the amplification process, the amplification and bandwidth narrowing takes place even if there were no cavity. Therefore, as the gain increases, the peak frequency gets even stronger and the line width becomes narrower.

Even if the line narrowing mechanism is already known, the steady-state linewidth of a laser is still difficult to predict. To the best of our knowledge, the gain saturation and steady-state bandwidth in NRLs have not been

explored yet. Unlike in ASE, an NRL will amplify the light until a gain saturates and becomes equal to the cavity loss. When we simply applied the Lorentzian shape of $\sigma_{st}(\nu) = \sigma_{st}(\nu_0) / \left[1 + ((\nu - \nu_0) / \Delta\nu)^2 \right]$ to our laser model [Eqs. (6) and (7) of the manuscript], we found the model expects *zero* steady-state bandwidth regardless of pumping power, which does not explain the measurements. As the gain for the center frequency is equalized to the cavity loss by saturation, the gain for the other frequency is always smaller than the loss and suppressed exponentially.

We think additional factors must be considered to correctly and fully explain and predict the steady-state linewidth of NRL. Fundamentally, we believe the effect of spontaneous emission must be considered to exactly predict the steady-state bandwidth of NRL similar to that of conventional lasers [Schawlow, Arthur L., and Charles H. Townes. *Physical Review* (1958)]. In many practical situations, however, we expect that the practical environments such as pumping power instability and thermal effect will define the lower bound of laser linewidth.

Based on the reviewer's comment, we inserted a discussion on the laser line narrowing mechanism in the revised manuscript.

5. What is the spatial profile of the output laser radiation? is it sensitive to the pump level?

Thank you for bringing our attention to this point. As suggested, we examined the spatial profile of the laser beam (Fig. R2). Because we used a coupling fiber to outcouple the laser emission, the spatial profile of the laser is confined to the 200 μm diameter core of coupling fiber. However, due to the used coupling lens ($f = 4.51 \text{ mm}$) and collimating lens ($f = 300 \text{ mm}$) (see Method for details), the final spatial profile of the laser is larger than 10 mm in diameter. Because the size is bigger than a typical image sensor, we placed an additional $4f$ system with a magnification factor of $\times 0.5$ to examine the spatial profile of the laser beam.

We found that the beam spatial profiles are independent of the pumping powers, as expected. The decrease in the visibility of Newton's rings is observed for the low pumping power as the laser linewidth broadens. Note that the interference effect is originated from the protection glass in the image sensor and is not the intrinsic property of the laser beam.

As suggested, the spatial profiles of output laser radiation in various pumping powers are added (Supplementary Figure 2) and discussed to the revised manuscript.

Figure R2. Beam spatial profiles in different pumping powers (P_I). The measured image is normalised to its mean value. Dotted circle extrapolates the boundary of beam profile.

6. It would be nice to illustrate the steady-state spatial distribution of the pump and laser power over the sphere surface estimated by the model.

Thanks for the suggestion. Regarding to the intensity distribution in the scattering cavity, we adopted the

calculations, which had been used for integrating spheres [Carr, K. F. *Surface coatings international* 80.8 (1997): 380-385]. Assuming the Lambertian refection, the probability that a photon leaves the surface to solid angle $d\Omega$ of angle θ (from the line that is normal to the surface) is,

$$p(\theta) = \frac{1}{\pi} \cos \theta d\Omega. \quad (\text{R1})$$

Then, the probability that a photon leaves dA_1 and arrives dA_2 in the spherical cavity is (see Fig. R3),

$$p(\theta) = \left(\frac{1}{\pi} \cos \theta \right) \left(\frac{1}{S^2} \cos \theta dA_2 \right) = \frac{1}{\pi D_{cav}^2} dA_2, \quad (\text{R2})$$

which is independent of the scattering angle θ . This means that every point on the spherical scattering cavity illuminates all other points at the very same probability. Based on this calculation, we can assume the uniform intensity distribution on the cavity internal wall.

Figure R3. A schematic of the spherical scattering cavity, brought from the Fig. 2 of [Carr, K. F. *Surface coatings international* 80.8 (1997): 380-385].

Still, the initial spatial profile of pumping light may become dominant as the internal area fraction of aperture increases or the wall reflectivity for pumping light decreases. In order to minimize such effects, we distributed the pumping light to the cavity surface by coarsely grinding the internal fiber end.

As suggested, the internal intensity distribution is clarified in the revised manuscript.

7. The reflection from the fiber face is not taken into account for the results shown in Fig. 5. Could you please comment importance of this effect? Does this reflection higher or lower than the reflection from the cavity walls?

Thank you for the comment. The reflection from the fiber face was already considered in Fig. 5 of the original manuscript (or Fig. 6 of revised manuscript). The R_{oc} of equation (S1) of revised manuscript considers the back-reflection from fiber ends and fiber impurities [the reflectance of the case (ii) in Fig. 6b], and the R_{noc} of the same equation considers the back-reflection from fiber ends and the surface of cylindrical channel [the reflectance of the case (iii) in Fig. 6b]. The fiber end reflectance can be estimated by using Fresnel equations and the refractive index (RI) difference between the fiber (RI ≈ 1.45) and air (RI = 1) and the calculated reflectance is 0.034 for the normal angle, which is usually much smaller than the reflectance of cavity wall (> 0.9).

Since R_{oc} and R_{noc} affect both the numerator and denominator of equation (S2) of the revised manuscript, the fiber end reflection is an important factor in the lasing efficiency calculation. However, in practical situations, the first term of equation (S1) of the revised manuscript usually becomes dominant, because the scattering wall area is significantly larger than the aperture area ($p_{wall} \gg p_{oc}, p_{noc}$) and the reflectance of the cavity wall is bigger than that of fiber end ($R_{wall} > R_{oc}, R_{noc}$). In such cases, the increasing fiber end reflection will monotonically reduce the laser slope efficiency due to the $(1 - R_{oc})$ term in the numerator of the equation (S1).

As the reviewer recommended, we added the discussion on the effect of fiber end reflection in the revised manuscript.

What is the mechanism preventing intracavity lasing?

Thank you for the question. Since non-resonant feedback does not require specific laser cavity geometry or phase relations, our system can lase whenever the sufficient pumping power is provided. However, we find too big size of a cavity will require an impractically high threshold for a pumping power. For example, if one uses a cavity of the same material with $D_a = 4.35$ mm and $D_{cav} = 10$ mm ($\beta = 2.3$), the expected pumping threshold is >200 W according to the Fig. 6d of the revised manuscript, which is impractical in typical lab situations. Based on the reviewer's comment, we inserted an additional discussion about the required laser threshold in the revised manuscript.

Reviewer #2 (Remarks to the Author):

Report on “Non-resonant power-efficient directional Nd:YAG ceramic laser using scattering cavity” by KyeoReh Lee *et al.* In this paper, KR Lee *et al.* investigate non-resonant laser based on a spherical cavity with a small entrance. Using a small-aperture spherical cavity inside porous ND3+: YAG ceramic medium, laser-like emission is observed with significant enhancement in the slop efficiency and linewidth. By adjusting the size of the outlet and diameter of the spherical cavity, the emission properties of the system were tuned. A theoretical model was also developed to elucidate and predict the lasing behaviors of the proposed non-resonant laser. The results seem to be correct, and they are clearly presented. There is no doubt that the paper deserves to be published. However, I am not sure that there is enough novelty to meet the publication criteria for Nature Communications. The presented results are somewhat expected. I also find some important details are missing in the manuscript.

We appreciate the careful reading and the constructive comments.

1. The author claims that the proposed non-resonant light trap design leads to power-efficient directional lasing emission. However, in the manuscript, no information about the emission direction is provided.

Thank you for the comment. We quantified the directionality of the proposed laser based on the number of participating spatial modes (N_x) in the Result [Snyder, Allan W., and John Love. *Optical waveguide theory*. Springer]. As the smaller N_x presents the better directionality of laser. The N_x of the proposed system is calculated from the properties of a coupling fiber. We found $N_x \approx 21,800$, which is 10^2 - 10^3 folds smaller than the N_x of common random lasers ($> 10^7$). Nevertheless, we admit that the relation between N_x and directionality and the enhancements in directionality were not clearly stated in the original manuscript. We revised and clarified the related paragraph accordingly in the revised manuscript.

2. The whole paragraph on page 5, “The smooth transition at the threshold ... effectively lifts the power efficiency” is unclear.

We appreciate the comment. We agree the previous manuscript was not written in a clear way. The manuscript has been revised accordingly.

The kink at the threshold lasers originated from the different radiation properties between laser and spontaneous emission. Unlike the spontaneous emission usually radiates in every direction with the intrinsic spectrum of a gain medium, the laser emission usually radiates in a specific direction with a sharp spectral shape defined by the lasing mode. Therefore, the “sharpness” of the kink at laser threshold can be quantified by the fraction of the spontaneous emission that coupled with the lasing mode, and is called spontaneous emission coefficient β factor (please do not confuse with the symbol used in our manuscript $\beta = D_{cav}/D_a$) [Yoshihiso Yamamoto, and Richard E. Slusher *Physics Today* 46, 6, 66 (1993); van Soest, Gijs, and Ad Lagendijk. *Physical Review E* (2002)]. The β factor ranges from 0 to 1. The bigger β factor presents the smoother the kink at the laser threshold. For $\beta = 1$ the kink is completely vanished (see Fig. 2 of Yamamoto *et al.* for details).

In conventional lasers, the spontaneous emission coefficient β factor is usually close to zero, which exhibits sharp kink. However, some ‘thresholdless lasers’ suppress the occupiable optical modes of spontaneous emission by carefully designed microcavities. In that way, the β factor can be lifted, even up to $\beta = 1$ [Yoshihiso Yamamoto, and Richard E. Slusher *Physics Today* 46, 6, 66 (1993)].

Conversely, the β factor can be increased by the inflation of laser emitting modes. For example, in many random lasers, laser emission direction covers the entire 4π solid angle, which is identical to that of spontaneous emission. In such random lasers, therefore, the β factor is usually determined only from the spectral overlap between the laser and spontaneous emission. The β factor of random lasers is usually 0.01 – 1, which is much bigger than the β factor of conventional lasers [van Soest, Gijs, and Ad Lagendijk. *Physical Review E* (2002)].

Our laser system is similar to the random laser situations. Both laser and spontaneous emission generated in the cavity can only escape through the coupling fiber. Even if the possible number of lasing spatial modes (N_x) is

hugely suppressed, the occupiable spatial mode is still identical to that of spontaneous emission. Therefore, we cannot differentiate them in a spatial domain. In the work, we used an emission filter ($\lambda_c = 1064$ nm, $\Delta\lambda_{\text{FWHM}} = 10$ nm) to differentiate the laser emission from the spontaneous emission. If our power measurement covers the entire emission spectrum range of Nd:YAG including the secondary transitions such as ${}^4F_{3/2} \rightarrow {}^4I_{9/2}$ (939-946 nm) and ${}^4F_{3/2} \rightarrow {}^4I_{13/2}$ (1319-1357 nm), the kink may not be observable.

3. The cooling scheme to reduce the thermal effects is not properly discussed in the manuscript.

Thank you for the comment. We added the details on the cooling system on the Method section of the revised manuscript.

4. Figures 4c and 4c are not referred to in the text.

Thank you. This mistake was fixed in the revised manuscript.

5. It is not clear how the scattering gain medium amplifies the light upon each reflection and prevents extinction.

Thanks for the comment. For clarity, let us split your question in twofold: (i) how does the scattering gain medium amplify the light upon each reflection, and (ii) how does the scattering gain medium prevent extinction (of the emission light)?

To the question (i), because the scattering medium (Nd:YAG) is a gain medium that is sufficiently excited by the pumping light, the emission light will experience amplification by the stimulated emission during scattering events. This is the same process that happened in random lasers. Regarding the question (ii), since the amplification compensates for the reflection losses, the emission light effectively can live long enough to find the exit.

Based on the reviewer's comment, we also clarified this point in the revised manuscript.

6. Some crucial information like the scattering mean free path of the porous ceramic medium, the penetration depth of the pump light, and the surface roughness of the inner wall of the cavity are missing in the manuscript.

Thanks for the comment. As commented, we have presented the scattering properties of the used materials in the Experimental Methods section of the revised manuscript.

The absorption and reduced scattering coefficients (the inverse of reduced scattering mean free path) for pumping and emission wavelengths (804 and 1064 nm) were estimated in 'Fabrication of Nd:YAG porous bulk ceramic' of Method section. The effective dwell times and penetration depths of both pumping and emission light were also discussed in Supplementary Method 2.

We think that the *surface roughness of scattering medium* is not an important parameter in our system. In the scattering cavity, the reflected light from the scattering wall will be scrambled regardless of surface roughness. The reviewer may intend to consider the portion of specular reflection. But in our system, the portion of specular reflection is also independent of the surface roughness since the light incident angle to wall surface is already randomly distributed in the cavity.

Reviewer #3 (Remarks to the Author):

A random laser does not necessarily need a cavity, but some designs have one. Here is an example:

<https://www.nature.com/articles/srep35225>. In that design, a gain medium is surrounded by a scattering surface (by the way, I think that it would be appropriate to cite that paper).

We appreciate a helpful reference. We found the given reference is closely related to our work. We cited the work in the revised manuscript.

As far as I understand, the main idea behind the manuscript I'm reviewing is to invert that concept and make an empty cavity surrounded by a gain medium, where the cavity surface scatters light. A fiber couples the pump light into this hollow cavity and also couples the emitted light out of it. As far as I know, this is a new idea. Unfortunately, I do not have the necessary background to judge how good the idea is. Neither the slope efficiencies nor the pumping thresholds reported here look outstanding to me, but I am not an expert in random lasers, so I do not want to draw any conclusions from this fact.

We sincerely appreciate the careful reading and the comment.

I would just like to encourage the authors to clarify whether their design outperforms other random lasers in any particular respect. Whether it does or does not, it should be written explicitly.

We appreciate the constructive comment. The present concept resolved the issues of low power efficiency and omnidirectional emission – the major limitations of conventional random lasers. To clearly demonstrate the comparison between our work and conventional random lasers, we showed the random laser result without the cavity (the gray lines in Figs. 3a, 3c, and 3d). Our results show that the power efficiency and directionality are enhanced by the factor of 25 and 10^2 - 10^3 . We also find the linewidth narrowing effect is increased more than 10 times. As suggested, we have clarified these points in the revised manuscript.

My biggest problem with this manuscript is its theoretical part. I find it so incomplete that I cannot fully understand what the authors did. I think I understand the concept of effective reflectance for the pump light -- it must be the ratio of the power that gets reflected back into the fiber to the power that is sent into the fiber. But what is the effective reflectance for the emitted light? Also, I do not understand Eq. (S2) even for the pump light. If a pump photon enters the cavity via the fiber, it will get reflected from a wall for sure. Then the probability that it will enter the fiber again depends on what happens next: whether the photon gets into the surrounding gain medium or hits the wall again at another place or hits the fiber. I just do not see how Eq. (S2) reflects all this. If Eq. (S2) is correct, the authors must explain it in a detailed and careful way, so that even a reader like me, who apparently lacks some relevant knowledge, can understand the very basics of their model. The same applies to Eq. (S1).

Thanks for the helpful comment, which we fully agree with. As suggested, in the revised manuscript, we have thoroughly revised the theoretical parts (Supplementary Information).

The effective reflectance in the manuscript (R_{eff}^p or R_{eff}^{em}) presents the mean probability that the light reflects back to the intracavity space whenever light encounters the internal surface of a cavity. It can be calculated by the sum of (*statistical probability*) \times (*reflectance*) = $\sum_k p_k R_k$ of (i) the wall reflection, (ii) output coupling, (iii) loss through the aperture, and (iv) undesired leaks (Fig. 6b of the revised manuscript). This is how equation (S2) [or equation (S1) of the revised manuscript] is constructed. Notice the effective reflectance for pumping and emission light are different because the wall reflectance in two different wavelengths $R_{wall}^p = 0.913$ and $R_{wall}^{em} = 0.989$ are different.

Similarly, the total chance of loss for a single intracavity reflection event is expressed as the sum of (*statistical probability*) \times ($1 - \text{reflectance}$) = $\sum_k p_k (1 - R_k) = \sum_k p_k - \sum_k p_k R_k = 1 - R_{eff}$. Then, the pumping

efficiency (η_p) can be defined as the fraction of wall absorption among the total loss, (*wall absorption chance*) / (*total loss chance*) = $p_{\text{wall}} (1 - R_{\text{wall}}^p) / (1 - R_{\text{eff}}^p)$. The output coupling efficiency (η_{oc}) can also be calculated in a similar way, (*outcoupling chance by the fiber*) / (*total loss chance*) = $p_{oc} (1 - R_{oc}) / (1 - R_{\text{eff}}^{em})$. This is how equation (S1) [or equation (S2) of the revised manuscript] is constructed.

The authors highlight the perspective of new lasers in the ultraviolet and terahertz frequency regions as potential further developments based on their approach. Here, I find that the authors owe the community a couple of simple estimations in the Supplementary Materials. Based on their theoretical understanding, do they have any evidence that their design offers advantages for ultraviolet or terahertz lasing, as compared to other random lasers?

Thanks for the comment. Our perspective on the ultraviolet (UV) and terahertz (THz) frequency regions are based on the existing random lasers realized in the region of UV [Cao, Hui, et al. *Physical Review Letters* 82.11 (1999): 2278.] and THz [Schönhuber, Sebastian, et al. *Optica* 3.10 (2016): 1035-1038]. Since our design and theory can be applied to any scattering material, we believed the same concept and geometry can readily applicable to the different frequency regimes by appropriately choosing a scattering gain medium.

Based on the theory we proposed, the slope efficiency and threshold are highly dependent on the wall reflectance of a scattering medium, which is again closely related to the scattering properties of the scattering medium (such as absorption and scattering coefficients). In order to estimate the possibilities and feasibilities of different NRLs, therefore, we need to first estimate the scattering properties of a scattering gain medium.

Although making assumptions on the scattering properties of a certain material is possible in principle, such an approach would not reflect all the experimental situations. This is because, the scattering properties of a material are not fundamental parameters of the given material, but are indeed very sensitive to the manufacturing process. For example, the same Nd:YAG medium can become turbid, translucent, and transparent depending on the fabrication method and the consequent microscopic structures of the material. For this reason, even if we can provide certain analysis by assuming the scattering properties, the feasibility of the estimated properties cannot be guaranteed until the actual demonstrations, and the corresponding discussions and conclusions may hugely be misled. Based on this reason, providing the direct evidence that our design offers advantages for ultraviolet or terahertz lasing is not feasible at this point. In the revised manuscript, we have toned down for the further applications, which can be good topics for follow-up studies.

A few minor remarks:

- I think that the first equation on line 419 is wrong -- the first denominator should be 4 instead of 2.

Thanks for the comment. The equation is $A_{\text{tot}} = \frac{\pi}{2} D_{\text{cav}}^2 (1 + \cos \theta) + \frac{\pi}{4} D_a^2$, where $\theta = \sin^{-1}(D_a / D_{\text{cav}})$. We find the $\pi/2$ coefficient is correct for the first term. The equation calculates the total inner surface of our cavity. The first term is the area of spherical cap excluding the area that reserved by the aperture. For examples, if there is no aperture ($D_a = 0$), the first term becomes πD_{cav}^2 , which is the full inner surface area of a sphere, and if $D_a = D_{\text{cav}}$, the first term becomes $\frac{\pi}{2} D_{\text{cav}}^2$, which is the half inner surface area sphere (please see schematics of $\beta = 1$ in Fig. 6c of revised manuscript).

- Please check what you wrote on line 179.

We agree the previous expression was not clear. The manuscript is revised as: "... *practical parameters such as reflection from the fibre ($R_{oc} = 0$), loss through aperture ($p_{noc} = 0$), and the unwanted leaks in cavity ($p_{leak} = 0$). ...*"

- In Figure 2b, I do not see any "semicylinder channel" with a diameter of 250 micrometers.

We added a magnified image in Fig. 2b in the revised manuscript.

In summary, my overall impression of this manuscript is positive, I find its main idea novel and appealing, and if the authors can highlight in what respect their approach outperforms other random lasers, I recommend the manuscript for Nature Communications. Otherwise, if it is just a nice idea that has some potential, then a journal like Scientific Reports is probably better suited for disseminating these results. In any case, the authors should heavily revise their manuscript for clarity, especially its theory part.

We appreciate the all comments of the reviewer. We believe the revised manuscript now clearly highlights the better power efficiency and directionality compared with conventional random lasers based on the experimental results. The theoretical part is revised thoroughly in order to deliver clearer ideas to the readers. Especially in the Supplementary Method, the equation derivation process is elucidated step-by-step. We hope the revised manuscript has resolved all issues that the reviewer identified.

REVIEWERS' COMMENTS

Reviewer #1 (Remarks to the Author):

The authors' revision improves the manuscript quality.
The presented new explanations and expressions seem to be reasonable
The paper deserves publications.

Reviewer #2 (Remarks to the Author):

The author replied satisfactorily to the concern raised by me. Therefore, the paper can be accepted for publication in Nature Communications.

Reviewer #3 (Remarks to the Author):

The authors have appropriately revised their manuscript, and I recommend it for publication.

Reviewer #1 (Remarks to the Author):

The authors' revision improves the manuscript quality.
The presented new explanations and expressions seem to be reasonable
The paper deserves publications.

We appreciate the referee for their valuable time and the contribution to the manuscript.

Reviewer #2 (Remarks to the Author):

The author replied satisfactorily to the concern raised by me. Therefore, the paper can be accepted for publication in Nature Communications.

We appreciate the referee for their valuable time and the contribution to our manuscript.

Reviewer #3 (Remarks to the Author):

The authors have appropriately revised their manuscript, and I recommend it for publication.

We appreciate the referee for their valuable time and the contribution to our manuscript.